# Use, characteristics and influence of lay consultation networks on treatment-seeking decisions in slums of Nigeria: a cross-sectional survey

Chinwe Onuegbu [1], Jenny Harlock,[1] Frances Griffiths [1,2]

¹Division of Health Sciences, Warwick Medical School, University of Warwick, Coventry, UK
²Centre for Health Policy, University of the Witwatersrand, Johannesburg, South Africa

**Correspondence to**
Chinwe Onuegbu;
chinwe.onuegbu@warwick.ac.uk

## ABSTRACT

**Objectives** To describe the use, characteristics and influence of lay consultants on treatment-seeking decisions of adults in slums of Nigeria.

**Design** Cross-sectional survey using a pre-piloted questionnaire.

**Settings** Two slum communities in Ibadan city, Nigeria.

**Participants** 480 adults within the working age group (18–64).

**Results** Most respondents (400/480, 83.7%) spoke to at least one lay consultant during their last illness/health concern. In total, 683 lay consultants were contacted; all from personal networks such as family and friends. No respondent listed online network members or platforms. About nine in 10 persons spoke to a lay consultant about an illness/health concern without intending to seek any particular support. However, almost all (680/683, 97%) lay consultants who were contacted provided some form of support. Marital status (OR=1.92, 95% CI: 1.10 to 3.33) and perceiving that an illness or health concern had some effects on their daily activities (OR=3.25, 95% CI: 1.94 to 5.46) had a significant independent association with speaking to at least one lay consultant. Age had a significant independent association with having lay consultation networks comprising non-family members only (OR=0.95, 95% CI: 0.92 to 0.99) or mixed networks (family and non-family members) (OR=0.97, 95% CI: 0.95 to 0.99), rather than family-only networks. Network characteristics influenced individual treatment decisions as participants who contacted networks comprising non-family members only (OR=0.23, 95% CI: 0.08 to 0.67) and dispersed networks (combination of household, neighbourhood and distant network members) (OR=2.04, 95% CI: 1.02 to 4.09) were significantly more likely to use informal than formal healthcare, while controlling for individual characteristics.

**Conclusions** Health programmes in urban slums should consider engaging community members so, when consulted within their networks, they are able to deliver reliable information about health and treatment-seeking.

## INTRODUCTION

The health of people living in slums is a global health priority.[1] About 60% of urban dwellers in sub-Saharan Africa live in slums.[1] Slums are low resource and overcrowded environments

## STRENGTHS AND LIMITATIONS OF THIS STUDY

⇒ To the best of our knowledge, this is the first survey to extensively describe the use, characteristics and influence of lay consultation networks on management of illnesses and health concerns in slum settings.

⇒ The study was conducted among adults recruited through a robust sampling strategy in slum communities which are hard-to-reach communities for researchers.

⇒ This study used an established social network research technique for identifying egocentric networks. This is a novel application and makes a unique contribution to the literature on slum health in low-income and middle-income countries.

⇒ The data supplied by respondents about their network members were not independently verified by interviewing the network members.

⇒ There may have been recall bias as participants were asked to recall information about their illness concerns and use of networks, however the short recall period minimises this risk.

deficient in health-related infrastructures, including water, sanitation and hygiene, adequate housing structure and secured tenure.[2] These conditions increase physical and mental health risks. Slum dwellers are exposed to negative neighbourhood effects that affect individual health beyond personal or household characteristics.[3 4] For example, formal and comprehensive healthcare facilities are scarce or inaccessible due to cost or distance, which contributes to the use of healthcare providers such as drug shops and alternative health practitioners.[5 6] Reliance on social support from informal sources such as personal network members are common in these contexts.[7]

Lay consultants are the personal network members (eg, family and friends) or informal online sources/networks (eg, websites, Facebook groups) that individuals discuss an illness or perceived health concern with.[8 9]

People speak to lay consultants for three main reasons: (1) to casually report a health concern without intending to seek support, (2) to consult lay consultants for advice and information and (3) to seek instrumental forms of support, for example, loans.[10] People are likely to engage different network members for specific reasons.[11] Some avoid discussing their health concerns with others for personal reasons, for example, to maintain privacy.[12] Lay consultants provide various forms of support, including information or advice, appraisal of symptoms/health problems, instrumental support (eg, cash) and emotional support (eg, listening).[13] Provision of these resources has positive or negative consequences on personal treatment decisions.[14] For instance, lay consultants encourage or discourage people from seeking care from a formal healthcare provider.[15 16]

Network characteristics including network size (number of people consulted per illness episode), composition (characteristics of the group, eg, proportion of kin) and culture (eg, groups preference for formal healthcare practitioners) are associated with individual treatment-seeking decisions.[14 17] The characteristics of a person's network are associated with the person's sociodemographic factors such as age, gender, marital status, residence and access to healthcare.[13 18–21] For instance, women are more likely than men to speak to more people and have diverse networks per illness episode.[22 23] These dynamics are important to understand how people make treatment decisions.

Slum dwellers face socioeconomic and environmental conditions that shape their personal and online networks. Being employed in informal labour with tight working conditions and frequent migration in slums contributes to small, closely knit networks, and limits networking with wider networks.[24 25] Living in clustered and intimately shared environments creates avenues for easy interaction,[26] but contributes to loss of individual privacy and friction among neighbours.[27] Digital gaps and low digital literacy is common among slum dwellers in low-income and middle-income countries (LMIC),[28] which will likely affect access to online lay networks for advice.

There is empirical evidence suggesting slum dwellers interact with personal network members including family, friends and neighbours during illness and obtain a range of resources.[5 7 29 30] A recent systematic review demonstrated that lay consultation have positive and negative consequences on treatment seeking behaviours in slum settings of LMICs .[31] However, there are evidence gaps on network characteristics, how personal networks and online networks are combined and how the network characteristics impact on personal health seeking decisions. Getting this evidence requires egocentric techniques, which are used in network studies to elicit information about a network from an ego (individual); this is lacking in slum studies.[7] Thus, this study responds to these evidence gaps and calls for more population-based studies from informal settlements of LMICs.[32 33]

## Research objectives
In the study conducted in slums in Nigeria, we focus on three objectives:
► To describe the use and characteristics of lay consultation networks (personal and online).
► To describe the factors associated with the use and composition of lay consultation networks.
► To examine the association between characteristics of lay consultation networks and individual's decision to use formal or informal health services.

## METHODS
### Study design
#### NIHR slum health project: surveys undertaken prior to this study
This survey builds on a multicountry project: the NIHR Global Research Unit on Improving Health in Slums (the slum health project).[34] The slum health project involved household surveys across seven slums in Nigeria, Kenya, Bangladesh and Pakistan, to investigate access to and use of healthcare services. The household survey in Nigeria was conducted in 2017, in one slum in Lagos state and two slums in Oyo state.[35] Maps of the structures in the sites were generated and used to select samples for the household survey.[34] Findings from the Nigerian household survey can be accessed here.[35]

The survey reported in this article builds on the NIHR slum health survey in the two slum areas in Oyo state, Nigeria.

We adopted a cross-sectional design and conducted the survey between November 2020 and January 2021. We used an egocentric survey design to elicit data about networks. An egocentric survey involves using a name generator to ask a participant (ego) about their network members (alters) and the relationship between them and their network members.[36] This differs from a sociocentric study where information is obtained from egos and alters.[36] Both designs are common in network studies and are selected based on the research question. For this study on people's perception of their use of lay consultation networks, an egocentric design was appropriate.

The survey was followed by qualitative research.[37]

### Setting and participants
The survey was conducted in the two slum sites located in Ibadan, Oyo state. The sites are anonymised to protect their identities. The slums were selected as they met the definition of urban slums, and were accessible based on negotiations by the prior NIHR slum health survey.[34] Ibadan is one of the largest indigenous cities in Africa.[38] The city has an annual population growth rate of around 2.39% and a current population of about 3 million.[38 39] Slums in Ibadan are characterised by poor housing quality, tenure insecurity, insecurity of lives and property and the absence of basic infrastructures, including water supply and proper housing structures.[40 41] One of the slums surveyed was a migrant community of around 5800 persons, most of whom are young and involved in trading. The second slum was an indigenous slums inhabiting

around 5500 persons. Both slums have a high presence of informal healthcare providers including Patent Medicine Vendors and traditional/spiritual healers.[35]

We recruited a sample of 480 adults between 18 and 64 years residing in the communities. The criteria for inclusion included being aged 18 to 64, and residing in the community. We used the Cochran sample size formula (p=0.5, CI=95%) with an attrition rate of 25%, to arrive at a sample size of 480. We recruited equal numbers of males and females across three age groups (18–29, 30–49 and 50–64) to ensure age/sex representation. To recruit the participants, we obtained a spatially referenced sampling frame listing all the structures/buildings in the two study communities, and randomly selected 480 structures. We then assigned a unique identifier and participant category (eg, male 18–29 and female 18–29) to each structure. Fieldworkers used Global Positioning System to locate each structure to recruit the assigned participant category. In structures with more than one household, fieldworkers listed the households and randomly selected one to participate in the study. Similarly, when a household had more than one person who fit the participant category required, the persons were allotted unique identifiers and a lottery system was used to select one person. Where the assigned participant category was not found in the structure, the structure was recorded and automatically reassigned to another participant category. During the recruitment, three people decided not to take part, and they were replaced with three other participants.

### Data collection

We collected data using a semi-structured questionnaire (see online supplemental table 1). The survey instrument contained questions adapted from previously validated questionnaires, including the Nigeria Demographic Health Survey (NDHS, 2018),[42] the slum health project[34] and Pew research centre (2013).[43] We added questions developed from reviewing the literature.[23] The instrument comprised five parts with questions on:

1. Sociodemographic status of respondent.
2. A recent illness experience/health concern for which the respondent needed care or advice (open-ended question), their interactions with lay network members or lay online sources of health advice and types of healthcare used for this recent experience/concern.
3. Name-generators[36]—respondents asked to list all the lay network members or lay online sources of health advice for the recent experience/concern.
4. Sociodemographic characteristics, reasons for the interaction and support obtained from each name mentioned in 3.
5. General use of online platforms for advice seeking.

The questionnaire was translated into two local languages (Yoruba and Hausa) by linguists and then back-translated by different linguists to check consistency. Following this, it was reviewed at a community workshop in each field site, involving six community members selected through snowballing. We reworked the questionnaire based on their comments, and pretested it with a convenience sample of 20 participants residing just beyond our field sites. We analysed the responses using simple frequencies and percentages, and reviewed the questionnaire based on the results.

Trained fieldworkers administered the questionnaire using the Open-Data Kit (ODK), an open-source software tool that enables data collection, storage and analysis using mobile android devices. Respondents could choose to participate in the studies face to face or through mobile calls. The COVID-19 public health guidelines provided by the Nigerian government were strictly followed for face-to-face data collection.

### Data analysis

The questions on the recency of an illness or health concern experienced by respondents, nature of illness or health concern and how the illness or health concern affected people's daily activities/functioning were open-ended questions. Responses were first content analysed and then put into categories.

We calculated frequencies and simple percentages for all measures. We conducted bivariate analyses using $\chi^2$ to explore the association between network members individual characteristics with type of conversation they were consulted for, and type of support they provided.

We ran multivariable associations using a series of logistic regression models

1. individual characteristics and use of at least one lay consultant
2. individual characteristics and having non-kin or mixed networks than family only networks
3. network characteristics and use of formal health services while controlling for individual characteristics.

The logistic model analyses were performed at a 95% CI limit, and a p value less than 0.05 was deemed statistically significant. All statistical analysis was done using SPSS V.24.0.

### Patient and public involvement

We involved residents and stakeholders from the communities in reviewing the instruments prior to the survey. Findings from this survey will be shared with residents and local lawmakers in the communities.

## RESULTS
### Sample characteristics

A total of 480 participants (240 per site) took part in the survey. Response rate was very high at 99.4% (attrition rate 0.6%, 3/480). Across the data set, 98% of all questions were completed were completed. The sociodemographic characteristics of the study sample (see table 1) was very similar to that of the population of the informal settlements as established in the slum health study.[28 44]

Just under half of the sample reported they experienced an illness/health concern in the week preceding the survey, and that they felt that they need needed care

**Table 1** Sociodemographic characteristics, last illness/health concern for which advice or care was needed and use of healthcare services

| Variable | n=480 | (%) |
|---|---|---|
| **Sociodemographics** | | |
| Age categories | Mean 37.8, SD 14 | |
| 18–29 | 176 | 36.7 |
| 30–49 | 166 | 34.6 |
| 50–64 | 138 | 28.8 |
| Gender | | |
| Male | 223 | 46.5 |
| Female | 257 | 53.5 |
| Marital status | | |
| Married or living together | 310 | 64.6 |
| Not married | 170 | 35.4 |
| Employment status | | |
| Currently employed | 367 | 76.5 |
| Not currently employed | 113 | 24.5 |
| Migration history | | |
| Born in the neighbourhood | 133 | 27.7 |
| Moved from elsewhere | 347 | 72.3 |
| Daily mobile phone access | | |
| Yes | 432 | 90 |
| No | 48 | 10 |
| Internet access | | |
| At least once weekly | 211 | 44 |
| Never | 269 | 56 |
| Health insurance | | |
| Yes | 18 | 3.7 |
| No | 462 | 96.3 |
| **Last illness/health concern experience** | | |
| Last illness/health concern experience | n=480 | |
| Within the last week | 225 | (46.9) |
| Within the last month | 136 | (26.3) |
| Within the last 3 months | 54 | (11.3) |
| More than 3 months ago | 75 | (15.6) |
| Nature of illness or health concern | n=479* | |
| Infections (malaria, typhoid) | 196 | 40.8 |
| Headaches | 72 | 15.0 |
| General pain and weakness | 65 | 13.5 |
| Specific musculoskeletal pain | 49 | 10.2 |
| Gastrointestinal and abdominal issues | 29 | 6.0 |
| Others | 69 | 14.4 |
| Effect of symptom of health concern in daily functioning/activities | n=480 | |

Continued

**Table 1** Continued

| Variable | n=480 | (%) |
|---|---|---|
| Some effects | 266 | 55.4 |
| No effects | 214 | 44.6 |
| **Healthcare utilisation** | | |
| Sought care or treatment | n=480 | |
| Yes | 450 | 93.8 |
| No | 30 | 6.2 |
| **Type of health service used** | n=450 | |
| **Formal healthcare services** (medical drs/registered nurse/physiotherapists) | 107 | 23.8 |
| **Non-formal care:** patent medicine vendor | 209 | 46.4 |
| Self-medication | 43 | 9.6 |
| Pharmacist | 32 | 7.1 |
| Auxiliary nurse | 27 | 6 |
| Home remedy | 17 | 3.8 |
| Traditional medicine practitioner | 13 | 2.9 |
| Spiritualist | 2 | 0.4 |

*One non-response was recorded.

or advice for it. Of these, 196 (40.8%) reported illness of health concern was an infection (144 specified malaria). Other illness/health concerns were headaches, general pain and weakness, specific musculoskeletal pain, gastrointestinal and abdominal issues and 'others' (please see online supplemental table 2 for list of illnesses reported). Almost all the respondents used some form of care, and majority of them used informal healthcare practices/services. (see table 1).

### Use and characteristics of lay consultation networks

Most respondents spoke to at least one lay consultant during their last illness/health concern. All the lay consultants used were from personal networks; no respondent listed an online network or platform. About nine in 10 persons reported conversations in which they discussed an illness/health concern without intending to seek any particular support. However, almost all the respondents obtained some form of support following the conversations. Lay consultation networks mostly comprised one to two persons, family members only and members of the same household or neighbourhood. The lay consultants included spouses/partners (204/683, 29.9%), friends/neighbours 135/683, 19.7), children (131/683, 19.2%), parent (124/683, 18.2%), other relatives (76/683, 11.2%), coworkers (10/683, 1.5%) and clergy (3/683, 0.4%).

Just over half of the sample had networks where no one had internet access. Only very few respondents (27/480, 5.6%) said they had ever used online sources for seeking health advice (see table 2). There was a significant association between relationship to network members and

**Table 2** Use and characteristics of lay consultation networks, and general use of online sources for advice-seeking

| Variables | No | % |
|---|---|---|
| **Use of lay consultants** | | |
| Discussion with at least one person (online or offline) or platform | n=480 | |
| Yes | 400 | 83.7 |
| No | 80 | 16.3 |
| **Characteristics of lay consultation/lay consultant** | | |
| Relationship to all lay consultants* | n=683 | |
| Kin | 529 | 77.5 |
| Non-kin | 154 | 22.5 |
| Kind of relationship to lay consultant | n=683 | |
| Spouse/partner | 204 | 29.9 |
| Friend/neighbour | 135 | 19.7 |
| Child | 131 | 19.2 |
| Parent | 124 | 18.2 |
| Other relative | 76 | 11.2 |
| Coworker | 10 | 1.5 |
| Clergy | 3 | 0.4 |
| Gender of all lay consultants* | n=683 | |
| Male | 294 | 43 |
| Female | 389 | 57 |
| Means of communication with all lay consultants* | n=683 | |
| Face to face | 597 | 87 |
| Mobile phone call | 89 | 13 |
| Reason for contacting lay consultant* | n=683 | |
| Reporting | 590 | (86.4) |
| Consulting | 84 | (9.3) |
| Instrumental | 9 | (1.3) |
| Support obtained from lay consultants* | n=683 | |
| Information | 277 | (40.7) |
| Appraisal | 35 | (6.0%) |
| Instrumental | 327 | (48.1) |
| Emotional | 41 | (5.1) |
| No support | 3 | (0.4) |
| **Structure of lay consultation networks** | | |
| Lay consultation network size | n=480 | |
| 0 | 80 | 16.7 |
| 1 | 197 | 41 |
| 2 | 142 | 29.6 |
| 3 | 46 | 9.6 |
| 4 | 11 | 2.3 |
| 5 | 4 | 0.8 |
| Relationship composition | n=400 | |

Continued

**Table 2** Continued

| Variables | No | % |
|---|---|---|
| Family only | 279 | (69.8) |
| Non-family only | 45 | (11.3) |
| Mixed (family and non-family) networks | 76 | (19.0) |
| Gender composition | n=400 | |
| Women only | 153 | (38.3) |
| Men only | 106 | (26.5) |
| Mixed (women and men) | 141 | (35.3) |
| Geographic location of network members | n=400 | |
| Household only | 197 | (49.3) |
| Neighbourhood only | 34 | (8.5) |
| Household or neighbourhood | 46 | (11.5) |
| Distant networks (located beyond the household or neighbourhood) | 47 | (11.8) |
| Dispersed networks (combination of household, neighbourhood and distant network members) | 76 | (19.0) |
| Proportion of networks that used the internet | n=400 | |
| At least one network member had internet access | 195 | 48.8 |
| None of the network members had network access | 205 | (51.2) |
| Medicine preference in the network | n=400 | |
| All promote modern medicine only: | | |
| Yes | 170 | 42.5 |
| No | 230 | 57.5 |
| **General use of online sources for advice seeking** | | |
| Ever used an online source for advice seeking | 27 | 5.6 |
| Never used an online source for advice seeking | 453 | 94.4 |
| If yes, which do you use?† | n=27 | |
| Google | 19 | 70.4 |
| Facebook | 10 | 37.0 |
| YouTube | 4 | 14.8 |
| In what ways do you share such health concerns/symptoms on online platforms?† | | |
| Browsing | 22 | 81.5 |
| Posting health-related questions on health professional sites or social media platforms | 4 | 14.8 |
| Posting health-related questions on platforms of persons personally known to you | 4 | 14.8 |
| Posting health-related questions on sites/platforms with more general audience | 1 | 3.7 |

Continued

**Table 2** Continued

| Variables | No | % |
|---|---|---|
| What do you post or share online?† | | |
| Specific question about your health | 4 | 14.8 |
| Others- browse symptoms | 23 | 85.2 |

*This is all the lay consultants contacted: 400 respondents contacted 683 lay consultants.
†Percentage does not total 100 because it is a multichoice question.

type of conversations held with them $\chi^2$ (1, n=683)=50.8, p<0.001). Respondents were significantly more likely to engage in consulting conversations (in which they are asking for advice or information) with non-family members than family members (29.2 vs 7.4). Similarly, we found a significant association between relationship to network members and type of support obtained from them $\chi^2$ (3, n=683)=16.5, p<0.001). Non-family members were more likely than family members to provide information (52.3% vs 37.4%). A slightly higher proportion of non-family members than family members provided appraisal of symptoms (7.3% vs 4.5%) and emotional support (6.6% vs 5.9%). In contrast, family members were more likely than non-family members to give instrumental support (52.2% vs 33.8%). These results are presented in online supplemental tables 3 and 4.

### Factors associated with the use and composition of lay consultation networks

Being married (OR=1.92; 95% CI: 1.10 to 3.33) and perceiving that an illness or health concern had some effects on daily activities (OR=3.25; 95% CI: 1.94 to 5.46) had a significant independent association with speaking to at least one lay consultant (see table 3, model 1). Age had a significant independent association with speaking to non-family only networks (OR=0.95; 95% CI: 0.92 to 0.99)

**Table 4** Multivariable analyses exploring the association between network variables and use of formal healthcare while controlling for age, gender, employment, marital status, Internet access, health insurance and perceiving that an illness affects daily activities

| Network variables | Use of formal care OR (95% CI) |
|---|---|
| Non-family only networks | 0.23 (0.08 to 0.67)** |
| Mixed (family and non-family) networks | 0.96 (0.50 to 1.86) |
| Women only networks | 0.89 (0.46 to 1.71) |
| Mixed gender networks | 0.87 (0.42 to 1.82) |
| Distant networks | 1.95 (0.87 to 4.35) |
| Dispersed networks | 2.04 (1.02 to 4.09)* |
| All network members promote formal health services | 1.01 (0.62 to 1.66) |
| At least one network member has internet | 1.23 (0.74 to 2.05) |

*P<0.05, **p<0.01, ***p<0.001.
†F

and mixed (family and non-family) networks (OR=0.97; 95% CI: 0.95 to 0.99) than family-only networks (see table 3, models 2 and 3, respectively).

### Association between network characteristics and use of health services

Table 4 shows that network variables influenced healthcare seeking, independent of personal characteristics. Respondents who interacted with non-family members only (OR=0.23, 95% CI: 0.08 to 0.67) were less likely to use formal healthcare instead of informal healthcare than those who interacted with family-only networks and mixed networks. Those who interacted with network members in dispersed locations (OR=2.04, 95% CI: 1.02

**Table 3** Multivariable logistic regression results: relationship between respondents characteristics and having at least one lay consultant, non-family only networks, and mixed networks

| Respondents characteristics | At least one lay consultant Model 1 OR (95% CI) | Non-family only networks† Model 2 OR (95% CI) | Mixed networks† Model 3 OR (95% CI) |
|---|---|---|---|
| Age | 0.99 (0.97 to 1.01) | 0.95 (0.92 to 0.99)** | 0.97 (0.95 to 0.99)** |
| Female | 1.06 (0.63 to 1.79) | 1.76 (0.85 to 3.61) | 0.62 (0.35 to 1.09) |
| Currently employed | 0.55 (0.28 to 1.10) | 0.65 (0.28 to 1.54) | 1.26 (0.70 to 2.27) |
| Currently married | 1.92 (1.10 to 3.33)* | 1.44 (0.64 to 3.25) | 0.75 (0.40 to 1.42) |
| Access to the internet at least once weekly | 0.74 (0.41 to 1.33) | 0.73 (0.37 to 1.47) | 1.07 (0.58 to 1.97) |
| Access to mobile phones daily | 1.27 (0.53 to 3.05) | 0.49 (0.10 to 2.38) | 0.46 (0.16 to 1.27) |
| Access to health insurance | 4.93 (0.62 to 38.98) | 0.36 (0.09 to 1.45) | 1.07 (0.58 to 1.97) |
| Perceived that an illness/health concern affected daily activities | 3.25 (1.94 to 5.46)*** | 0.77 (0.38 to 1.56) | 0.65 (0.38 to 1.13) |

*P<0.05, **p<0.01, ***p<0.001.
†Family-only network is the reference group.

to 4.09) were more likely than those that spoke to household/neighbourhood and distant networks to use formal care instead of informal healthcare.

## DISCUSSION

This study demonstrated that respondents commonly interacted with lay consultants when they perceived an illness or health concern. The lay consultation networks used were small and dominated by personal network members characterised by face-to-face contacts. While we intended to describe both personal and online networks, we found that no respondent used online sources of lay advice and people rarely used them when managing illnesses or health concerns.

The finding that most people with illness symptoms or health concerns spoke to at least one network member before seeking formal or informal care is unsurprising. Similar findings have been highlighted in other low-income,[45] and high-income contexts globally,[13 15 23] reiterating that lay consultation is a common process of responding to illness across different context.[46] In low resource settings of LMICs including slums, speaking to network members about illnesses or health concerns may be driven by limited access to formal health advice sources such as official symptom checkers or government health websites.[47] Reliance on informal sources of health for advice and support is common in slums, and it is linked to shortage of public healthcare facilities and inadequate capacity to pay for the private healthcare.[48]

Small lay consultation networks dominated by family members have been found in non-slum settings of LMICs and high-income settings.[17 21] Health is a sensitive issue, and people are likely to speak to small number of persons who are available to them and whom they can trust to provide the support they need.[49] In slum settings, it is not unusual for people of working age to discuss their concerns with small number of persons, as they are likely to be engaged with busy work life which makes networking challenging.[25] Keeping small networks of trusted persons in slums is also linked to frequent migration, low reciprocity among neighbours to avoid being overburdened by the needs of others and lack of trust.[24]

Our finding that married respondents were likely to speak to a lay consultant supports previous evidence that spouses are important contacts in people's treatment pathway.[19 50] Respondents who thought a health concern affected their daily activities were likely to speak to a consultant. This is in line with published evidence that people are likely to initiate treatment-seeking decisions, part of which is speaking to lay consultants, for symptoms that affect their vocational and social activities.[51] Absence from work have particularly undesirable for slum dwellers who engage in daily informal labour for a daily wage,[5] which makes it unsurprising that they engaged in lay consultation about illness concerns they perceived to affect their daily activities. As reported in other studies,[17 52] older persons were less likely to

have non-family only networks or mixed networks. This relates to the socioemotional selectivity theory that an increase in age is associated with smaller social network sizes and network members that provide emotional support.[53]

Although our respondent's lay consultation networks were dominated by family members, they were more likely to specifically seek advice from non-family members than family members. This supports the functional specificity theory,[54] which suggests that people seek specific forms of support from their different networks. Our findings showed that non-family members were more likely than family members to provide information, while family members were more likely than non-family to provide instrumental support. Similarly, other studies suggest that friends and networks beyond one's immediate household are more likely to be sources of new information and ideas and family members are more likely to give hands-on support.[55 56] Our findings revealed that respondents obtained appraisal of symptoms and emotional support from slightly more non-family members than family members. This is line with published evidence that show that peers are important for evaluating and guiding others with similar symptoms and providing emotional support.[23 57]

Our finding that participants who interacted with non-kin ties and extensive network members (combination of household/neighbourhood/distant network members) were significantly more likely to use informal healthcare rather than formal healthcare is similar to what has been found in other non-slum contexts in a high-income country.[58] Patent Medicine Vendors (also known as Chemists) were the most commonly used informal healthcare among our study participants, corroborating findings from other similar studies.[35 48] While information and suggestions about medicines and remedies can be retrieved from family members, non-kin ties and extensive networks can provide information about new and unfamiliar options.[58]

Our survey found that none of the participants utilised online sources in their recent illness/health concern experience and that online sources were scarcely used overall. This contrasts with findings from research conducted in high-income countries which indicated that internet sources were commonly incorporated into people's lay consultation network.[9 59 60] The contrast might be due to the digital and literacy gaps among slum dwellers in LMICs.[28]

Moreover, the survey findings highlight the concept of network homophily, whereby people tend to associate with those with whom they share similar characteristics.[61] We found that both participants and their network members had limited internet access. In low resource settings, individuals within networks face similar constraints which affect how they can support each other or share resources.[62] This reinforces the social exclusion and marginalisation experienced by people living in slum environments.

This survey had some strengths and limitations. Our survey was conducted among adults in slum settings, which are hard-to-reach communities for researchers.[24] We used a robust sampling approach, which allowed us to capture data from a representative sample of adults from 18 to 64 years. We asked about use of both online and personal network members to gain a holistic view of people's lay consultation network in the digital era.

This study inherited the limitations of self-reported egocentric network studies in which the data supplied by respondents were not independently verified.[63] For instance, while respondents reported that a network member did not have access to the internet, the data were not verified by interviewing the network member. However, participants' perception of the characteristics of their lay consultants is an important factor for understanding why people engage in lay consultation.[14] There may have been recall bias as participants were asked to recall information about their illness concerns and use of networks, however the short recall period minimises this risk.

In summary, people living in urban slums interact with their informal network members to make treatment-seeking decisions. During history taking, healthcare practitioners can investigate advice that people have received and tried, to understand why patients seek their help and other expectations.[64] Health programmes (such as health campaigns) in urban slums should consider engaging community members so, when consulted within their networks, they are able to deliver reliable information about health and treatment-seeking. The findings from this study are likely to be transferable to other slums of LMICs with similar socioeconomic characteristics, limited access to comprehensive and quality formal care, and dominance of informal healthcare.[4 5 48]

**Contributors** CO conducted the data analysis and wrote the entire article. JH and FG were involved in the design of the research, reviewed and revised the paper. All authors approved the final version of the paper. CO is reponsible for the overall content as guarantor.

**Funding** This research was funded by the National Institute for Health Research (NIHR) Global Health Research Unit on Improving Health in Slums using UK aid from the UK Government to support global health research. Grant reference number 16/136/87.

**Disclaimer** The views expressed in this publication are those of the author(s) and not necessarily those of the NIHR or the UK Department of Health and Social Care.

**Competing interests** None declared.

**Patient and public involvement** Patients and/or the public were involved in the design, or conduct, or reporting or dissemination plans of this research. Refer to the Methods section for further details.

**Patient consent for publication** Not required.

**Ethics approval** This study involves human participants. All participants provided informed consent to participate before taking part in the study. Ethical approval was obtained from the Biomedical and Scientific Research Ethics Sub-Committee, University of Warwick, UK (BSREC 17/19-20 AM01) and the Research Ethics Committee of the Oyo State Ministry of Health, Nigeria (AD13/479/2035). Participants gave informed consent to participate in the study before taking part.

**Provenance and peer review** Not commissioned; externally peer reviewed.

**Data availability statement** Data are available upon reasonable request. The data are available from the study authors on request.

**ORCID iDs**
Chinwe Onuegbu http://orcid.org/0000-0001-6372-9390
Frances Griffiths http://orcid.org/0000-0002-4173-1438

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
