## [Reviewer comments · BMJ Open]

ARTICLE DETAILS

TITLE (PROVISIONAL)	Use, characteristics, and influence of lay consultation networks on treatment-seeking decisions in slums of Nigeria: a cross-sectional survey
AUTHORS	Onuegbu, Chinwe; Harlock, Jenny; Griffiths, Frances

VERSION 1 – REVIEW

REVIEWER	Rephaim Mpfu University of Cape Town, Department of medicine
REVIEW RETURNED	09-Aug-2022

GENERAL COMMENTS	This manuscript reports findings from a study that aimed to assess characteristics associated with the use of networks to assist in the management of health concerns or illnesses. A cross-sectional study design was employed, and structures within slums in Nigeria were sampled. Overall, they found that the majority of those sampled had consulted a lay consultant before, and that significant factors that affected the manner and type of network included marital status, current employment status, age, and others. The manuscript was well written, concept is interesting and novel, and has important implications for improving healthcare in the considered populations. The results are well presented, but there is some information omitted that could improve the manuscript. Important omissions from the main report include strengths and limitations of the study. It is noted that qualitative work has been done and reported by the same research team, however, I have not seen it cited in the background and study rationale, which should be considered if relevant. Please see specific comments below: • Abstract: Include absolute numbers when quoting percentages• "Marital status ($P < 0.05$, OR 1.92, 95% CI 1.10, 3.33) and perceiving that an illness or health concern had some effects on their daily activities ($P < 0.001$, OR 3.25, 95% CI 1.94, 5.46) had a significant independent association with speaking to at least one lay consultant." Please re-organize these statistics. It should be ordered by relevance, consider the following: point estimate, confidence interval, p-value). Also, consider using different delimiters for the different statistics vs the confidence interval band delimiters, i.e. use a different punctuation mark for the confidence interval vs separation of stats • Strengths and limitations:o Did you examine online based lay consultants? The results indicate that there was no response and so data were not reportedo The use of name generators is not discussed at all in the paper. Consider removing as a strength, or revise paper to discuss this point.
---

	 o Strengths and limitations not discussed in the manuscript • Results: Please describe some of the recruitment data such as attrition rate or instances where random selection of households/household participants was required • Consider sharing the questionnaire, for critical appraisal as well for potential use in future studies and increasing generalizability • Results: report the data for some of the associations found, rather than just reporting their conclusions, e.g. "from the multivariable modelling we found marital status and perceiving that an illness or health concern had some effects on their daily activities has (sic) a significant independent association with speaking to at least one lay consultant" can include a quotation of the point estimate and uncertainty interval. • Tables: Indicate what this range represents. 95% CIs? • Consider a brief discussion on why respondents may prefer to consult a lay consultant rather than accessing formal health care, as that may have interesting application for improving healthcare access pathways in the sample population • Tables: Consider reporting all p-values rather than only selecting statistically significant results
--	---

REVIEWER	Edgar Arnold Lungu UNICEF Zambia
REVIEW RETURNED	23-Jan-2023

GENERAL COMMENTS	Dear Sir/Madam Thank you for the opportunity to review this manuscript. The authors tackle an important issue in urban health and the influence of lay consultants in care seeking decisions. I present below my review and comments which the authors should consider Background  - Check the grammar on line 19-20 indicating "...this contexts" - Check grammar on line 51 "influence" Methodology  - Page 5; Lines 4 to 10 are not clear. A survey was conducted in 2017 and then there is another that indicates of a survey between November 2020 and January 2021 - Under the setting and participants a bit more detail would be useful. For example  o What was the rationale and/or criteria for choosing the two slum areas in Oyo State o The participant eligibility criteria could be more explicit – is it just age (adult) and implicitly residence in selected area and house o If population figures exist, it would be useful to share these to obtain broader context of the slum size Results  - Lines 19 to 22 where it indicates response rate was very high, it would be advisable to share the exact response rate. The fact that those declined were replaced by others was in my view only a means to still reach the sample size. There is still need to indicate response rate (those who consented to respond relative to all requested to) - Lines 28 to 34 just before table 1, might need further thinking through. It doesn't look like there was any method to ascertain the type of illness so am curious how the authors conclude that 'infections were the common type' given that the participants arguably have no expertise to undertake diagnoses - Lines 28 to 34 – how was need defined here?. There is need for clarity on this. Sentence indicates "Just under half of the sample reported an illness/health concern that needed care or advice". Is
--

	this just all people who reported having a health concern/illness or there was a further filter of how many of those that had an illness actually needed care. If it is the latter then how was need determined? Perceived need from the respondent?  - Some sections of Table 2 might need further clarity or review on whether the table is the best way of presenting some of the information. For example:  o Not so sure of the utility of a variable “Proportion of networks that used the internet” It appears to me like it is presenting results on both column and rows. Use of internet is probably a good one where the options are Yes or No o Under medicine preference – what is the difference between “Mixed medical culture” and “All promote african traditional medicine and modern medicine” - Given this is a journal paper, multivariate results are probably worth of reporting as findings than bivariate for the methods section gives the impression that bivariate analysis was part of building the analytical model. Besides at this level of bivariate analysis, confounders are not controlled for so reporting such results may not be worthwhile for this purpose - For a multi-variate regression analysis, results under network variables are presented independently. Was network treated as one variable in the model Discussion  - Lines 16-20 suggest that findings are consistent with other evidence that spouses are an important contact in treatment pathways. There is no indication of that specific element in the results section. Whilst there is next of kin or family member, the results do not indicate anywhere that most often the lay consultant was actually a spouse. In essence lines 30-31 on page 14 indicate that the study did not ask respondents about the relationships between their network members - Whilst there are great discussion points, the discussion section could have benefited from a little more synthesis which combined reference to studies with similar and contrasting findings (which is well done) as well as reference to the context (what issues from the policy or study environment) explain the underlying dynamics and could explain the findings - The authors could have reflected on the strengths and limitations of the study more explicitly. In some ways lines 26 to 31 highlight some strengths but could be organized better and also reflect on limitations. For example recall bias is an obvious limitation given that people were asked to recall lots of information including their illness, symptoms and networks – however the short recall period minimizes this. The fact that generalizability of the findings may be an issue given the study setting could also have been highlighted
--	---

VERSION 1 – AUTHOR RESPONSE

Reviewer: 1

Overall comment:

This manuscript reports findings from a study that aimed to assess characteristics associated with the use of networks to assist in the management of health concerns or illnesses. A cross-sectional study design was employed, and structures within slums in Nigeria were sampled. Overall, they found that the majority of those sampled had consulted a lay consultant before, and that significant factors that affected the manner and type of network included marital status, current employment status, age, and others. The manuscript was well written, concept is interesting and novel, and has important

implications for improving healthcare in the considered populations. The results are well presented, but there is some information omitted that could improve the manuscript. Important omissions from the main report include strengths and limitations of the study. It is noted that qualitative work has been done and reported by the same research team, however, I have not seen it cited in the background and study rationale, which should be considered if relevant. Please see specific comments below:

Authors revision:

Thank you for taking time to review our article and providing comments to help us improve it. We have now discussed the strengths and limitations of the study (see page 16, line 15-26):

This survey had some strengths and limitations. Our survey was conducted among adults in slum settings, which are hard-to-reach communities for researchers.(24) We used a robust sampling approach, which allowed us to capture data from a representative sample of adults from 18-64 years. We asked about use of both online and personal network members to gain a holistic view of people's lay consultation network in the digital era.

This study inherited the limitations of self-reported egocentric network studies in which the data supplied by respondents were not independently verified.(63) For instance, while respondents reported that a network member did not have access to the internet, the data was not verified by interviewing the network member. However, participants' perception of the characteristics of their lay consultants is an important factor for understanding why people engage in lay consultation.(14) There may have been recall bias as participants were asked to recall information about their illness concerns and use of networks, however the short recall period minimises this risk.

We have included the reference for our qualitative work on the reference list and cited it on page 5, line 20: The survey was followed by qualitative research (Onuegbu et al., 2022).

Abstract

Reviewers comment

Include absolute numbers when quoting percentages

Authors revision

Thank you for the suggestion. We have now revised as follows:

Most respondents (400/480, 83.7%) spoke to at least one lay consultant during their last illness/health concern (Page 2, line 7-8).

However, almost all (680/683, 97%) the lay consultants that were contacted provided some form of support (Page 2, line11-12).

Reviewers comment

"Marital status ($P < 0.05$, OR 1.92, 95% CI 1.10, 3.33) and perceiving that an illness or health concern had some effects on their daily activities ($P < 0.001$, OR 3.25, 95% CI 1.94,5.46) had a significant independent association with speaking to at least one lay consultant." Please re-organize these statistics. It should be ordered by relevance, consider the following: point estimate, confidence interval, p-value). Also, consider using different delimiters for the different statistics vs the confidence interval band delimiters, i.e. use a different punctuation mark for the confidence interval vs separation of stats.

Authors revision

Thank you. The abstract has been revised as follows (page 2, 12-22):

Marital status (OR = 1.92, 95% CI: 1.10- 3.33) and perceiving that an illness or health concern had some effects on their daily activities (OR = 3.25, 95% CI: 1.94 - 5.46) had a significant independent association with speaking to at least one lay consultant. Age had a significant independent association with having lay consultation networks comprising non-family members only (OR = 0.95, 95% CI: 0.92 - 0.99) or mixed networks (non-family and family members) (OR = 0.97, 95% CI: 0.95 - 0.99), rather than family-only networks. Network characteristics influenced individual treatment decisions as participants who contacted networks comprising non-family members only (OR=0.23, 95% CI: 0.08 - 0.67) and dispersed networks (combination of household, neighbourhood and distant network members) (OR = 2.04, 95% CI: 1.02 - 4.09) were significantly more likely to use informal than formal healthcare, while controlling for individual characteristics.

Strengths and limitations

Reviewers comment

Did you examine online based lay consultants? The results indicate that there was no response and so data were not reported

Authors revision

Thank you for this comment. We confirm that participants were asked about the use of online consultants (see questions 7 and 11 on page 1 of the questionnaire) however 0 participants reported using online consultants/platforms (see page 10, line 3-5). Therefore, a network analysis of their online networks/lay consultants could not be performed. In our discussion (see page 16, 4-14), we explained that our findings contrast with findings from high income countries as reported in our discussion and we have suggested that this may be because of internet access, poor digital literacy, and general low use of internet within the networks (network homophily).

Reviewers comment

The use of name generators is not discussed at all in the paper. Consider removing as a strength, or revise paper to discuss this point.

Authors revision

Thank you for noting this. We have revised the paper to discuss the point as follows (Page 5, 13-19):

We adopted a cross-sectional design and conducted the survey between November 2020 and January 2021. We used an egocentric survey design to elicit data about networks. An egocentric survey involves using a name generator to ask a participant (ego) about their network members (alters) and the relationship between them and their network members.(36) This differs from a sociocentric study where information is obtained from egos and alters.(36) Both designs are common in network studies and are selected based on the research question. For this study on people's perception of their use of lay consultation networks, an egocentric design was appropriate.

Reviewers comment

Strengths and limitations not discussed in the manuscript

Authors revision

We have now discussed the strengths and limitations of the study (see page 16, line 15-26):

This survey had some strengths and limitations. Our survey was conducted among adults in slum settings, which are hard-to-reach communities for researchers.(24) We used a robust sampling approach, which allowed us to capture data from a representative sample of adults from 18-64 years. We asked about use of both online and personal network members to gain a holistic view of people's lay consultation network in the digital era.

This study inherited the limitations of self-reported egocentric network studies in which the data supplied by respondents were not independently verified.(63) For instance, while respondents reported that a network member did not have access to the internet, the data was not verified by interviewing the network member. However, participants' perception of the characteristics of their lay consultants is an important factor for understanding why people engage in lay consultation.(14) There may have been recall

bias as participants were asked to recall information about their illness concerns and use of networks, however the short recall period minimises this risk.

Results

Reviewers comment

Please describe some of the recruitment data such as attrition rate or instances where random selection of households/household participants was required.

Authors revision

Thank you for your suggestion. We have revised the document as follows:

Response rate was very high at 99.4% (attrition rate was 0.6%, 3/480) (Page 8, 15-16)

In the methods section, we have added the following comments:

During the recruitment, three people decided not to take part, and they were replaced with three other participants (Page 6, 17-18)

Reviewers comment

Consider sharing the questionnaire, for critical appraisal as well for potential use in future studies and increasing generalizability.

Authors revision

We have now provided the questionnaire as a supplementary file

Tables

Reviewers comment

Indicate what this range represents. 95% CIs?

Authors revision

We have now indicated 95% CI in Tables 3 and 4 (pages 13 and 14, respectively)

Table 3: Multivariable logistic regression results: relationship between personal characteristics and selected network characteristics; relationship between network characteristics and personal use of formal healthcare

	At least one lay consultant	Non-family only networks^a	Mixed networks^a
Respondent's characteristics	Model 1	Model 2	Model 3
	OR (95% CI)	OR (95% CI)	OR (95% CI)
Age	0.99 (0.97-1.01)	0.95 (0.92-0.99)**	0.97 (0.95-0.99)**
Female	1.06 (0.63-1.79)	1.76 (0.85-3.61)	0.62 (0.35-1.09)

Currently employed	0.55 (0.28-1.10)	0.65 (0.28-1.54)	1.26 (0.70-2.27)
Currently married	1.92 (1.10-3.33)*	1.44 (0.64-3.25)	0.75 (0.40-1.42)
Access to the internet at least once weekly	0.74 (0.41-1.33)	0.73 (0.37-1.47)	1.07 (0.58-1.97)
Access to mobile phones daily	1.27 (0.53-3.05)	0.49 (0.10-2.38)	0.46 (0.16-1.27)
Access to health insurance	4.93 (0.62-38.98)	0.36 (0.09-1.45)	1.07 (0.58-1.97)
Perceived that an illness/health concern affected daily activities	3.25 (1.94 -5.46)***	0.77 (0.38-1.56)	0.65 (0.38-1.13)

^aFamily-only network is the reference group

*P < 0.05, ** P<0.01, ***P<0.001

Table 4: Multivariate analyses exploring the association between network variables and use of formal healthcare while controlling for age, gender, employment, marital status, internet access, health insurance and perceiving that an illness affects daily activities

Network variables	Use of formal care OR (95% CI)
Non-family only networks	0.23 (0.08-0.67)**
Mixed relationship networks	0.96 (0.50-1.86)
Women only networks	0.89 (0.46-1.71)
Mixed gender networks	0.87 (0.42-1.82)
Distant networks	1.95 (0.87-4.35)
Dispersed networks	2.04 (1.02-4.09)*
All network members promote formal health services	1.01(0.62-1.66)
At least one network member has internet	1.23 (0.74- 2.05)

^aFamily-only network is the reference group

*P < 0.05, ** P<0.01, ***P<0.001

Reviewers comment

Consider a brief discussion on why respondents may prefer to consult a lay consultant rather than accessing formal health care, as that may have interesting application for improving healthcare access pathways in the sample population

Authors revision

Thank you for your comment, we have revised the revised the discussion as follows (page 14, 10-18):

The finding that most people with illness symptoms or health concerns spoke to at least one network member before seeking formal or informal care is unsurprising. Similar findings have been highlighted in other low-income,(45) and high-income contexts globally,(13,15,23) reiterating that lay consultation is a common process of responding to illness across different context.(46) In low resource settings of LMICs including slums, speaking to network members about illnesses or health concerns may be driven by limited access to formal health advice sources such as official symptom checkers or government health websites.(47) Reliance on informal sources of health for advice and support is common in slums, and it is linked to shortage of public healthcare facilities and inadequate capacity to pay for the private healthcare.(48)

In the conclusion, we have revised as follows (page 19, 27-29, page 20 1-5)

In summary, people living in urban slums interact with their informal network members to make treatment-seeking decisions. During history taking, healthcare practitioners can investigate advice that people have received and tried, to understand why patients seek their help and other expectation.(64) **Health programmes (such as health campaigns) in urban slums should consider engaging community members so, when consulted within their networks, they are able to deliver reliable information about health and treatment-seeking. The findings from this study are likely to be transferable to other slums of LMICs with similar socioeconomic characteristics, limited access to comprehensive and quality formal care, and dominance of informal healthcare (4,5, 48).**

Reviewers comment

Tables: Consider reporting all p-values rather than only selecting statistically significant results

Authors revision

Thank you for your comment. Tables reporting regression results (table 3, 4 see page) each have a legend with the following information: *P < 0.05, ** P<0.01, ***P<0.001, which explains the significance level. Estimates without * indicate they have P values as > 0.05.

Table 3: Multivariable logistic regression results: relationship between personal characteristics and selected network characteristics; relationship between network characteristics and personal use of formal healthcare

	At least one lay consultant	Non-family only networks^a	Mixed networks^a
Respondent's characteristics	Model 1 OR (95% CI)	Model 2 OR (95% CI)	Model 3 OR (95% CI)

Age	0.99 (0.97-1.01)	0.95 (0.92-0.99)**	0.97 (0.95-0.99)**
Female	1.06 (0.63-1.79)	1.76 (0.85-3.61)	0.62 (0.35-1.09)
Currently employed	0.55 (0.28-1.10)	0.65 (0.28-1.54)	1.26 (0.70-2.27)
Currently married	1.92 (1.10-3.33)*	1.44 (0.64-3.25)	0.75 (0.40-1.42)
Access to the internet at least once weekly	0.74 (0.41-1.33)	0.73 (0.37-1.47)	1.07 (0.58-1.97)
Access to mobile phones daily	1.27 (0.53-3.05)	0.49 (0.10-2.38)	0.46 (0.16-1.27)
Access to health insurance	4.93 (0.62-38.98)	0.36 (0.09-1.45)	1.07 (0.58-1.97)
Perceived that an illness/health concern affected daily activities	3.25 (1.94 -5.46)***	0.77 (0.38-1.56)	0.65 (0.38-1.13)

^aFamily-only network is the reference group

*P < 0.05, ** P<0.01, ***P<0.001

Table 4: Multivariate analyses exploring the association between network variables and use of formal healthcare while controlling for age, gender, employment, marital status, internet access, health insurance and perceiving that an illness affects daily activities

Network variables	Use of formal care OR (95% CI)
Non-family only networks	0.23 (0.08-0.67)**
Mixed relationship networks	0.96 (0.50-1.86)
Women only networks	0.89 (0.46-1.71)
Mixed gender networks	0.87 (0.42-1.82)
Distant networks	1.95 (0.87-4.35)
Dispersed networks	2.04 (1.02-4.09)*
All network members promote formal health services	1.01(0.62-1.66)
At least one network member has internet	1.23 (0.74- 2.05)

^aFamily-only network is the reference group

*P < 0.05, ** P<0.01, ***P<0.001

Reviewer: 2

Dr. Edgar Arnold Lungu, UNICEF Zambia

Comments to the Author:

Dear Sir/Madam

Thank you for the opportunity to review this manuscript. The authors tackle an important issue in urban health and the influence of lay consultants in care seeking decisions. I present below my review and comments which the authors should consider.

Reviewers comment

Thank you for taking out time to review our manuscripts and provide us feedback. We have addressed each comment below.

Background

Reviewers comment

Check the grammar on line 19-20 indicating "...this contexts"

Authors revision

Thank you. This has now been revised:

Reliance on social support from informal sources such as personal network members are common in these contexts (Page 3, 10-11).

Reviewers comment

Check grammar on line 51 "influence"

Authors revision

Thank you. This has now been revised:

These characteristics are themselves influenced by sociodemographic factors, including age, gender, marital status, residence, and access to healthcare, all of which shape the use of personal networks. (Page 3, line 27-29)

Methodology

Reviewers comment

Page 5; Lines 4 to 10 are not clear. A survey was conducted in 2017 and then there is another that indicates of a survey between November 2020 and January 2021

Authors revision

Thank you for your comment, this has now been revised (page 5, 4-12):

NIHR slum health project: Surveys undertaken prior to this study

This survey builds on a multi-country project: the NIHR Global Research Unit on Improving Health in Slums(the slum health project).(34) The slum health project involved household surveys across seven slums in Nigeria, Kenya, Bangladesh, and Pakistan. The primary aim to investigate access to and use of healthcare services. The household survey in Nigeria was conducted in 2017, in one slum in Lagos state and two slums in Oyo state.(35) Maps of the structures in the sites were generated and used to select samples for the household survey.(34) Findings from the Nigerian household survey can be accessed here.(35)

The survey reported in this article builds upon the NIHR slum health survey in the two slum areas in Oyo state, Nigeria

Reviewers comment

Under the setting and participants a bit more detail would be useful. For example, What was the rationale and/or criteria for choosing the two slum areas in Oyo State

Authors revision

Thank you for your comment. This has now been revised as follows (Page 5, 22-29; page 6. 1-3)

The survey was conducted in the two slum sites located in Ibadan, Oyo State. The sites are anonymised to protect their identities. The slums were selected as they met the definition of urban slums, and were accessible based on negotiations by the prior NIHR slum health survey.(34) Ibadan is one of the largest indigenous cities in Africa.(38) The city has a an annual population growth rate of around 2.39% and a current population of about 3 million.(38,39) Slums in Ibadan are characterised by poor housing quality, tenure insecurity, insecurity of lives and property, and the absence of basic infrastructures, including water supply and proper housing structures.(40,41) One of the slums surveyed was a migrant community of around 5800 persons, most of whom are young and involved in trading. The second slum was an indigenous slums inhabiting around 5,500 persons. Both slums have a high presence of informal healthcare providers including Patent Medicine Vendors and traditional/spiritual healers.(35)

Reviewers comment

The participant eligibility criteria could be more explicit – is it just age (adult) and implicitly residence in selected area and house

Authors revision

We have included the following comment:

The criteria for inclusion included: being aged 18 to 64, and residing in the community. (Page 6, 4-5)

Reviewers comment

If population figures exist, it would be useful to share these to obtain broader context of the slum size

Authors revision

Thank you for your comments, we have revised as follows (Page 5, 22-29; Page 6, 1):

Ibadan is one of the largest indigenous cities in Africa.(38) The city has a an annual population growth rate of around 2.39% and a current population of about 3 million.(38,39) Slums in Ibadan are characterised by poor housing quality, tenure insecurity, insecurity of lives and property, and the absence of basic infrastructures, including water supply and proper housing structures.(40,41) One of the slums surveyed was a migrant community of around 5800 persons, most of whom are young and involved in trading. The second slum was an indigenous slums inhabiting around 5,500 persons.

Results

Reviewers comment

Lines 19 to 22 where it indicates response rate was very high, it would be advisable to share the exact response rate. The fact that those declined were replaced by others was in my view only a means to still reach the sample size. There is still need to indicate response rate (those who consented to respond relative to all requested to)

Authors revision

Thank you for your suggestion. We have revised the document as follows:

Response rate was very high at 99.4% (attrition rate was 0.6%, 3/480) (Page 8, 15-16)

In the methods section, we have added the following comments:

During the recruitment, three people decided not to take part, and they were replaced with three other participants (Page 6, 17-18)

Reviewers comment

Lines 28 to 34 just before table 1, might need further thinking through. It doesn't look like there was any method to ascertain the type of illness so am curious how the authors conclude that 'infections were the common type' given that the participants arguably have no expertise to undertake diagnoses

Authors revision Thank you for your comment. We have revised the manuscript to clarify the issue.

In the methods section we have added that:

The questions on the recency of an illness or health concern experienced by respondents, nature of illness or health concern, and how the illness or health concern affected people's daily activities/functioning were open ended questions. Responses were first content analysed and then put into categories. (see page 7)

We have included the table below into supplementary file, Table 2

Table 2: nature of illness of health concern, perceived effects of illness or health concern on daily activities

Nature of illness or health concern	N=480	%
Infectious symptoms		
Malaria	126	26.3
Malaria with headache/weakness/nausea/runny nose/ache	37	6.7
Typhoid	10	2.1
Measles-like rash	7	1.5
Malaria and typhoid	7	1.5
Malaria and ulcer	4	0.8
Typhoid and cough	1	0.2
Diarrhoea	2	0.4
Headache, cold and weakness	2	0.4
	196	40.8
Headaches		
Headache	55	11.5
Headache and body pain	5	1
Headache and weakness	3	0.8
Headache and fever	4	0.8
Headache and body ache	3	0.6

Headache and cold	1	0.2
Headache and catarrh	1	0.2
	72	15.0
General pain and weakness		
Body aches/body pain	34	6.3
Body weakness	25	5.2
Shortage of blood	1	0.2
Body pain and weakness	2	0.4
Tiredness and temperature	1	0.2
Weakness and dizziness	2	0.4
	65	13.5
Specific musculoskeletal pain		
Leg pain/arm pain/shoulder/waist/thighs	29	4.4
Joint pain/knee pain	12	2.5
Chest pain	4	0.6
Back pain/ upper/lower	4	0.4
	49	10.2
Gastrointestinal and abdominal issues		
Piles	5	1
Stomach pain/lower abdominal pain/diarrhoea	17	3.5
Stomach ulcer	2	0.2
Stomach pain and headache	5	1
	29	6.0
Others		
Cough	5	1
Fever/high temperature	17	3.5
Accident/injuries/bruises	3	2.3
Catarrh	8	1.5
Menstrual cramps	3	0.8
Eye pain	4	0.8
Toothache	4	0.8
Pregnancy symptoms	4	0.8
Cough and catarrh	3	0.6
Anxiety/depression/sadness	3	0.6
Cataract/cloudy vision	3	0.6
High blood pressure	2	0.4
Miscarriage	2	0.4
Stress	2	0.4
Arthritis	2	0.4

Mouth ulcer	1	0.2
Dry throat	1	0.2
Vagina bleeding following menstruation	1	0.2
Fever and high bp	1	0.2
Fever, pile, and dizziness	1	0.2
Hernia surgery	1	0.2
Stroke	1	0.2
Hypertension	1	0.2
Stiffness	1	0.2
Smelly burps	1	0.2
	69	14.4
Missing	1	0.2
Perceived effect of illness/health concern on daily activities	N=480	%
Some effects	266	55.4
No effects	214	44.6
Kinds of effects	N=266	%
Unable to go to work	82	30.8
Unable to perform normal household chores	59	22.2
Unable to work full time	30	11.3
Could not do any of my normal activities	39	14.7
Was on bed rest	24	9.1
Unable to perform religious activities	12	4.5
Could not exercise	8	3
Unable to sleep normally	4	1.5
Affected work and socialisation	4	1.5
Unable to socialise with friends	2	0.8
Hospitalised	1	0.4
Could not eat	1	0.4

Reviewers comment

Lines 28 to 34 – how was need defined here?. There is need for clarity on this. Sentence indicates “Just under half of the sample reported an illness/health concern that needed care or advice”. Is this just all people who reported having a health concern/illness or there was a further filter of how many of those that had an illness actually needed care. If it is the latter then how was need determined? Perceived need from the respondent?

Authors revision

Thank you for your comment, we have made the following revision (page 8, 20-21):

Just under half of the sample reported that they experienced an illness/health concern in the week preceding the survey, and they felt that they needed care or advice for it

Reviewers comment

Some sections of Table 2 might need further clarity or review on whether the table is the best way of presenting some of the information. For example: Not so sure of the utility of a variable “Proportion of networks that used the internet” It appears to me like it is presenting results on both column and rows. Use of internet is probably a good one where the options are Yes or No

Authors revision

Thank you for your comment. We have revised the table as follows (Table 2, page 12)

Proportion of networks that used the internet	n=400	
At least one network member had internet access	195	48.8
None of the network members had internet access	205	(51.2)

Reviewers comment

Under medicine preference – what is the difference between “Mixed medical culture” and “All promote African traditional medicine and modern medicine”

Authors revision

Thank you for your comment. We have revised the table as follows (Table 2, page 12):

All network members promote modern medicine only:	Yes	170	42.5
	No	230	57.5

Reviewers comment

Given this is a journal paper, multivariate results are probably worth of reporting as findings than bivariate for the methods section gives the impression that bivariate analysis was part of building the analytical model. Besides at this level of bivariate analysis, confounders are not controlled for so reporting such results may not be worthwhile for this purpose.

Authors revision

Thank you for suggestion. We have now reworked the results section to present the multi-variable results.

Reviewers comment

For a multi-variate regression analysis, results under network variables are presented independently. Was network treated as one variable in the model

Authors revision

Thank you for your comment. We have revised the table as follows (Table 4, page 13):

Table 4: Multivariate analyses exploring the association between network variables and use of formal healthcare while controlling for age, gender, employment, marital status, internet access, health insurance and perceiving that an illness affects daily activities

Network variables	Use of formal care OR (95% CI)
Non-family only networks	0.23 (0.08-0.67)**
Mixed relationship networks	0.96 (0.50-1.86)
Women only networks	0.89 (0.46-1.71)
Mixed gender networks	0.87 (0.42-1.82)
Distant networks	1.95 (0.87-4.35)
Dispersed networks	2.04 (1.02-4.09)*
All network members promote formal health services	1.01(0.62-1.66)
At least one network member has internet	1.23 (0.74- 2.05)

^aFamily-only network is the reference group

*P < 0.05, ** P<0.01, ***P<0.001

Discussion

Reviewers comment

Lines 16-20 suggest that findings are consistent with other evidence that spouses are an important contact in treatment pathways. There is no indication of that specific element in the results section. Whilst there is next of kin or family member, the results do not indicate anywhere that most often the lay consultant was actually a spouse. In essence lines 30-31 on page 14 indicate that the study did not ask respondents about the relationships between their network members

Authors revision

Thank you for your comment. We have included information on the relationships between respondents and network members in the results (Page 10, 9-11):

The lay consultants included spouses/partners (204/683, 29.9%), friends/neighbours 135/683, 19.7), children (131/683, 19.2%), parent (124/683, 18.2%), other relatives 76/683, 11.2%), co-workers 10/683, 1.5%) and clergy 3/683, 0.4%).

Table 1: individual characteristics of lay consultants used during the last illness episode

Kind of relationship to lay consultant	n=683	
Spouse/ partner	204	29.9
Friend/neighbour	135	19.7
Child	131	19.2
Parent	124	18.2
Other relative	76	11.2
Co-worker	10	1.5

Clergy	3	0.4
--------	---	-----

Reviewers comment

Whilst there are great discussion points, the discussion section could have benefited from a little more synthesis which combined reference to studies with similar and contrasting findings (which is well done) as well as reference to the context (what issues from the policy or study environment) explain the underlying dynamics and could explain the findings)

Authors revision:

Thank you for the correction, we have now made the following revision (page 14, 10-24, page 15, 1-2))

The finding that most people with illness symptoms or health concerns spoke to at least one network member before seeking formal or informal care is unsurprising. Similar findings have been highlighted in other low-income,(45) and high-income contexts globally,(13,15,23) reiterating that lay consultation is a common process of responding to illness across different context.(46) In low resource settings of LMICs including slums, speaking to network members about illnesses or health concerns may be driven by limited access to formal health advice sources such as official symptom checkers or government health websites.(47) Reliance on informal sources of health for advice and support is common in slums, and it is linked to shortage of public healthcare facilities and inadequate capacity to pay for the private healthcare. (48)

Small lay consultation networks dominated by family members have been found in non-slum settings of LMICs and high-income settings.(17,21) Health is a sensitive issue, and people are likely to speak to small number of persons who are available to them and whom they can trust to provide the support they need.(49) In slum settings, it is not unusual for people of working age to discuss their concerns with small number of persons, as they are likely to be engaged with busy work life which makes networking challenging.(25) Keeping small networks of trusted persons in slums is also linked to frequent migration, low reciprocity among neighbours to avoid being overburdened by the needs of others and lack of trust.(24)

On page 16 (4-14):

Our survey found that none of the participants utilised online sources in their recent illness/health concern experience and that online sources were scarcely used overall. This contrasts with findings from research conducted in high-income countries which indicated that internet sources were commonly incorporated into people’s lay consultation network.(9,59,60) The contrast might be due to the digital and literacy gaps among slum dwellers in LMICs.(28)

Moreover, the survey findings highlight the concept of network homophily, whereby people tend to associate with those with whom they share similar characteristics.(61) We found that both participants and their network members had limited internet access. In low resource settings, individuals within networks face similar constraints which affects how they can support each other or share resources.(62) This reinforces the social exclusion and marginalisation experienced by people living in slum environments..

Reviewers comment

The authors could have reflected on the strengths and limitations of the study more explicitly. In some ways lines 26 to 31 highlight some strengths but could be organized better and also reflect on limitations. For example recall bias is an obvious limitation given that people were asked to recall lots of information including their illness, symptoms and networks – however the short recall period minimizes this. The fact that generalizability of the findings may be an issue given the study setting could also have been highlighted

Authors revision:

We have now discussed the strengths and limitations of the study (see page 15, line 15-29):

This survey had some strengths and limitations. Our survey was conducted among adults in slum settings, which are hard-to-reach communities for researchers.(24) We used a robust sampling approach, which allowed us to capture data from a representative sample of adults from 18-64 years. We asked about use of both online and personal network members to gain a holistic view of people's lay consultation network in the digital era.

This study inherited the limitations of self-reported egocentric network studies in which the data supplied by respondents were not independently verified.(63) For instance, while respondents reported that a network member did not have access to the internet, the data was not verified by interviewing the network member. However, participants' perception of the characteristics of their lay consultants is an important factor for understanding why people engage in lay consultation.(14) There may have been recall bias as participants were asked to recall information about their illness concerns and use of networks, however the short recall period minimises this risk.

We have included a comment on the transferability of our findings to similar settings (page 16, 16-19):

The findings from this study are likely to be transferable to other slums of LMICs with similar socioeconomic characteristics, limited access to comprehensive and quality formal care, and dominance of informal healthcare.(4,5,48)

VERSION 2 – REVIEW

REVIEWER	Rephaim Mpofo University of Cape Town, Department of medicine
REVIEW RETURNED	23-Mar-2023
GENERAL COMMENTS	No further comments.